# Polyphenols from Food and Natural Products: Neuroprotection and Safety

**DOI:** 10.3390/antiox9010061

**Published:** 2020-01-10

**Authors:** Rui F. M. Silva, Lea Pogačnik

**Affiliations:** 1Research Institute for Medicines (iMed.ULisboa) and Department of Biochemistry and Human Biology (DBBH), Faculty of Pharmacy, Universidade de Lisboa, 1649-003 Lisbon, Portugal; rfmsilva@ff.ulisboa.pt; 2Department of Food Science and Technology, Biotechnical Faculty, University of Ljubljana, SI-1000 Ljubljana, Slovenia

**Keywords:** bioavailability, in vitro models, neuroprotection, neurotoxicity, polyphenols, toxicity

## Abstract

Polyphenols are naturally occurring micronutrients that are present in many food sources. Besides being potent antioxidants, these molecules may also possess anti-inflammatory properties. Many studies have highlighted their potential role in the prevention and treatment of various pathological conditions connected to oxidative stress and inflammation (e.g., cancer, and cardiovascular and neurodegenerative disorders). Neurodegenerative diseases are globally one of the main causes of death and represent an enormous burden in terms of human suffering, social distress, and economic costs. Recent data expanded on the initial antioxidant-based mechanism of polyphenols’ action by showing that they are also able to modulate several cell-signaling pathways and mediators. The proposed benefits of polyphenols, either as protective/prophylactic substances or as therapeutic molecules, may be achieved by the consumption of a natural polyphenol-enriched diet, by their use as food supplements, or with formulations as pharmaceutical drugs/nutraceuticals. It has also been proved that the health effects of polyphenols depend on the consumed amount and their bioavailability. However, their overconsumption may raise safety concerns due to the accumulation of high levels of these molecules in the organism, particularly if we consider the loose regulatory legislation regarding the commercialization and use of food supplements. This review addresses the main beneficial effects of food polyphenols, and focuses on neuroprotection and the safety issues related to overconsumption.

## 1. Introduction

Polyphenols are naturally occurring micronutrients that are present in plants as essential physiological compounds [1]. They comprise a wide family of molecules bearing one or more phenolic rings and are present in many food sources like wine, green tea, grapes, vegetables, red fruits, and coffee [2,3]. It is generally accepted that most polyphenols are potent antioxidants [4,5] and may also possess anti-inflammatory properties [6,7].

Those properties attracted the interest of researchers to polyphenols, and many studies highlighted their potential role in the prevention and treatment of various pathological conditions connected to oxidative stress and inflammation, like cancer and cardiovascular and neurodegenerative disorders, and also of pollutant-induced cell damage [8,9,10,11,12]. Moreover, those food products are relatively abundant in the human diet, and several foods and beverages can provide more than 1 mg of polyphenolic content per serving, as shown by the study of Pérez–Jiménez et al. (2010) based on data from the Phenol-Explorer database [13]. As an example, by using the same database, Godos et al. (2017) estimated that an Italian study population had a mean intake of approximately 660 mg of polyphenols per day, obtained from nuts, tea, coffee, cherries, citrus fruits, vegetables, chocolate, and red wine [14], all regular constituents of the Mediterranean diet [15], included in the list of the Intangible Cultural Heritage of Humanity by UNESCO. In fact, olive-oil components were described as relevant pharmacological molecules [16]. However, major dietary sources of polyphenols may vary depending on the traditional diets adopted in various countries, thus, in Northern and Eastern European countries, the main dietary sources of polyphenols are represented mostly by beverages, such as coffee and tea [17,18], while in Southern European and Mediterranean countries, important dietary sources may be nuts, olive oil, fruits, and vegetables [14,19].

Recently, the food industry also became interested in byproducts derived from plants and fruits due to their rich content of polyphenols, and potential use in functional foods or food supplements [20]. Although some phenolic compounds are specific to some fruits and plants, many polyphenols are present in several food sources [2], and many fruits and vegetables produce more than one particular type of phenol, being more or less rich in an assortment of those compounds [21].

Furthermore, the level of polyphenols in the same plant is not constant, but varies with, for example, crop and atmospheric conditions [20,21]. This variation of polyphenol content in what appears to be the same plant or fruit makes it difficult to assess the ingested amount by a particular person. All these facts have to be considered in the balance of the potential beneficial roles of polyphenols versus the possibility of intensified accumulation, safe consumption, and toxic effects.

Novel data expanded on the initial antioxidant-based mechanism of polyphenols’ action by showing that they are also able to modulate several cell-signaling pathways and mediators in a wide range of human pathologies. In a recent publication, Patel et al. (2019) revised the pharmacological applications of curcumin in several diseases [22], as well as a wide range of pleiotropic actions in the modulation of cell-signal molecules. In diabetes, tea polyphenols were able to reduce the senescence of glomerular mesangial cells by regulating the activity of miR-126/Akt-p53-p21 pathways [23]. The consumption of flavonoid compounds seemed to also have a beneficial effect on colon-cancer prevention by modulating lysosome enzymes, increasing the expression of apoptotic factors like Bax, Bcl2, and caspase-3 in cancer cells, and regulating cellular respiratory and mitochondrial enzymes [11]. A recent review also pointed out the beneficial role of dietary polyphenols quercetin and epigallocatechin gallate (EGCG) in the prevention and treatment of obesity, with important impact on the prevention of cardiovascular diseases. Quercetin appeared to modulate adipogenic pathways like the adenosine-5′-monophosphate (AMP)-activated protein kinase, and upregulate the levels of phosphorylated AMP-activated protein kinase (AMPK) and its substrate, acetyl-CoA carboxylase, in 3T3-L1 preadipocytes, while EGCG appeared to inhibit the proliferation and differentiation of 3T3-L1 preadipocytes in mature adipocytes by arresting the cell cycle [24]. However, it is still controversial whether or not these products can naturally increase intrinsic brain defenses and avoid, or at least reduce, the initial insults that lead to the neurodegenerative process.

## 2. Beneficial Effects of Polyphenols and Neuroprotection

Neurodegenerative diseases are typically characterized as pathological conditions where particular groups of neurons are damaged or lost, disturbing the normal function of the central nervous system, either by impairing cognitive functions, motor functions, or both. Many of those illnesses are commonly associated with aging, but it is currently known that neurodegeneration develops in a subclinical form over years, with neuronal death occurring progressively over a lifetime, much before the first clinical signs are noticeable. Current predictors indicate a continuous increase in dementia cases that, between 2005 and 2030, may reach about 50% of the aged population [25]. Numerous studies [26,27,28,29] have been dedicated to the cellular mechanisms for neurodegeneration in several pathological conditions, such as Alzheimer’s, Parkinson’s, and Huntington’s disease, as well as amyotrophic lateral sclerosis. However, there are currently no effective therapies available to treat such diseases besides symptom amelioration [30,31].

In spite of their specific pathways, many of those conditions share common mechanisms, such as neuroinflammation [32] and oxidative stress [29,33]. In fact, the possible role of reduced expression or imbalance of oxidative-stress regulatory genes in aging and neurodegeneration, as well as the possible protection by antioxidants, was already reviewed [34]. Therefore, any strategies that can delay or prevent the onset of the disease, conveying neuroprotection, may be as important as the ones designed to treat it. The notion that diet can have a crucial role as one of those strategies has recently been proposed, leading to several studies focused on the importance of nutritional consumption of natural products, as food itself or as food supplements, that may convey neuroprotection [35,36].

One of the first indications of biological activity from food-derived molecules was the discovery of the antibacterial properties of curcumin, published in *Nature* in the late 1940s by Schraufstatter and Bernt [37]. Other food polyphenols, particularly resveratrol, also attracted the attention of researchers, as suggested by the possible association between red-wine consumption in France and the low incidences of coronary heart disease [38]. This association could be explained by the antioxidative properties of food polyphenols, in this case resveratrol, that were also found to convey neuroprotective activity [38,39,40].

Several studies using polyphenols, particularly the ones from red wine or green tea [41], have focused on their neuroprotective role in most neurodegenerative diseases, like recently described neuroprotection by epigallocatechin gallate (EGCG) from amyloid-beta-mediated neurotoxicity [42]. These studies also showed the ability of EGCG to inhibit Bax and cytochrome c translocation and autophagic pathways by increasing LC3-II [43], and to modulate mitochondrial functions [44]. It was also demonstrated that EGCG is able to significantly cross the human blood–brain barrier (BBB) model and protect cortical cultured neurons from oxidative-stress-induced cell death [45]. In fact, recent studies suggested that some flavonoids are indeed able to reach the brain [46], and it is now important to clarify by which mechanisms they exert neuroprotection.

Further examples of polyphenols pinpointed as promising molecules in the prevention and treatment of neurodegenerative diseases can be found in the literature [21,47,48,49,50,51]. One of the most relevant examples is resveratrol, shown to have neuroprotective properties by decreasing microglia-induced neuroinflammation, protecting the brain against hypoxic–ischemic damage and ameliorating cognitive function in the Alzheimer’s disease model [38,39,40,52]. It also seems to decrease age-related cognitive decline and increase cognitive function through SIRT1 modulation that, among other important functions, seems to modulate the growth of dendrites and axons [53]. The neuroprotective effects of resveratrol go beyond the central nervous system, since it also seemed to reduce NFκB-mediated neuroinflammation and endoplasmic reticulum stress in an ischemia–reperfusion model of vasculitis peripheral neuropathy; this condition arises from an obstruction in the blood vessels supplying peripheral nerves due to inflammation and may be related to neuropathic pain [54]. Protein kinase C gamma was also described as another target for resveratrol and EGCG in a way that its activation is associated with neuroprotection [55]. It is, however, interesting to verify that, as expected, the beneficial roles of polyphenols are not all equal in intensity and vary among different food sources. For example, it was found in an Alzheimer’s rat model that better neuroprotection was achieved by supplementation with green tea than with black tea or red wine [56]. In a similar mouse model, pomegranate juice seemed to decrease amyloid deposition and improve behavior tests after food supplementation [10]. Interestingly, it was recently published that blueberry supplementation of rat food was able to reduce microglial inflammatory reaction due to the graft transplant [57], particularly in aged rats, and also to protect neural cells from oxidative stress and attenuate microglia activation [58]. A similar anti-inflammatory effect was observed in vitro after the lipopolysaccharide (LPS) stimulation of BV-2 cells [59], a microglia cell line. Exciting results were also achieved with the well-known curcumin in Parkinson’s disease [60], as well as for medicinal plants used in traditional medicine, like *Centella asiatica*, which were shown to reduce mitochondrial dysfunction and oxidative stress while improving cognitive function in an Alzheimer’s in vivo model [51].

Furthermore, a growing field of research illustrates the possibility for epigenetic modulation by dietary consumption of polyphenols, namely, on the modulation of pro- and anti-inflammatory microRNAs [61]. An example of those properties is neuroprotection via autophagy modulation in a prion disease model [43]. A recent review [62] highlighted the epigenetic modulation of curcumin, including the inhibition of DNA methyltransferases, regulation of histone modifications through regulation of histone acetyltransferases and histone deacetylases (HDACs), as well as regulation of microRNAs. The modulation of endothelial-cell inflammation through the epigenetic regulation of NF-κB target genes by EGCG, proposed by Liu et al. [9] as a beneficial agent against environmental pollutants’ vascular toxicity, may also have an important impact on the protection of BBB function in neurodegenerative diseases.

Interestingly, a relationship between proliferation in neurogenic niches and nutrition may also exist [63], as well as a relationship between neurogenesis impairment and neuroinflammation [64]. These findings raise the possibility that modulation of neuronal precursors’ niches may minimize the decline that could be associated with age, or the neurodegenerative disease itself, and constitute a promising field for further investigation.

In sum, the inclusion of phenolic compounds in the diet or their use as supplements, nutraceuticals, or pharmacological drugs, seems to be promising in the prevention of several different pathologies, namely, neurodegenerative diseases. An extensive list of such diseases (including depression, glutamate excitotoxicity, epilepsy, hearing, and vision disturbances, and neurodegenerative diseases), as well as in vitro, ex vivo, and in vivo studies that evaluated the action mechanisms of phenolic acids in those conditions, were reviewed by Szwajgier et al. [65]. Interesting studies were conducted on the general population exploring the association between dietary polyphenols and depressive symptoms leading to similar results, including the potential role of phenolic acids [66,67]. However, the overconsumption of polyphenols may raise safety concerns due to accumulation of high levels of these molecules in the organism, particularly if we consider the loose regulatory legislation regarding the commercialization and use of food supplements.

The proposed benefits of polyphenols, either as protective/prophylactic substances or as therapeutic molecules, may be achieved by the consumption of a natural polyphenol-enriched diet, and by their use as food supplements or formulation as pharmaceutical drugs/nutraceuticals [68]. It was also proved that the health effects of polyphenols depend on the consumed amount and on their bioavailability [69].

## 3. In Vitro Models to Evaluate Polyphenol Neuroprotection

### 3.1. Bioavailability

One of the main concerns for the use of dietary polyphenols in neuroprotection is their bioavailability, for some considered small [70], as well as the potentially toxic effects in high concentrations. For this reason, in vitro digestion models are widely used to study the structural changes, digestibility, and release of food components under simulated gastrointestinal conditions. The most frequently used biological molecules included in the digestion models are digestive enzymes (pancreatin, pepsin, trypsin, chymotrypsin, peptidase, α-amylase, and lipase), bile salts, and mucin; digestion temperature is 37 °C. The most commonly used models simulate the stomach and small-intestine digestion stages. The gastric phase is initiated by the addition of a pepsin solution, and adjustment of the pH to 2.0, following incubation at 37 °C in a covered shaking water bath for 1 h. The small intestinal phase is initiated by adjusting the pH to 5.3, followed by the addition of small intestinal enzyme solution (lipase, pancreatin, and bile salts). The final sample pH is adjusted to 7.5, followed by 2 h incubation at 37 °C. To find the destination of bioactive compounds (polyphenols) of the selected plant extracts, samples can be analyzed several times during the course of digestion by an HPLC/DAD system [71,72].

### 3.2. Blood–Brain Barrier (BBB) Transposition

Regardless of all neuroprotective properties determined in cell models, it is crucial to know whether the selected compounds reach the target organ to be effective. Most in vitro models use a “simplified” in vitro model that mimics the properties of the human BBB, composed of confluent monolayers of human brain microvascular endothelial cells (HBMEC) [73]. These cells can be cultured in a porous membrane (Transwell), establishing a dual-compartment model where the upper one mimics blood circulation, and the lower the brain parenchyma. The concentration of molecules in the lower compartment indicates brain availability properties [74,75,76].

### 3.3. Neurotoxicity vs. Neuroprotection

To evaluate the neuroprotective abilities of the polyphenol molecules (or their neurotoxic levels) several in vitro approaches can be followed. Briefly, primary neuron cultures from a rat brain could be used as a simple model, since these are considered to be the most susceptible to neurodegenerative insult [26,40,45]. However, the involvement of the inflammatory response elicited by glial cells in brain-injury processes is becoming increasingly evident. For this reason, microglia, the main partners in brain immune response and defense, and astrocyte cultures may be used [57,58,77,78,79]. Cell stress to mimic pathologic conditions, in control and polyphenol-treated cells, must be chosen according to a specific disease, for example, exposure to oxidative milieu [80], MPP+ (widely used Parkinson’s inducer) [50,81], Aβ-peptides [82], or bacterial lipopolysaccharide (LPS) [77] as neuroinflammation inducer. Cell death and respective mechanisms (apoptosis, necrosis, necroptosis, autophagy) can be evaluated, as well as parameters and pathways of oxidative damage, inflammatory response and epigenetic alterations [83,84,85,86]. Finally, it is paramount to validate the most promising in vitro data using in vivo models, like toxicologically treated [54,87] or disease-specific [51,56] rodents that could be fed with the original foods, extracts, or isolated bioactive polyphenol molecules. This is also fundamental in assessing the in vivo side effects of polyphenols, particularly in situations mentioned above, and to ascertain the safe usage of any proposed polyphenol [88,89,90,91]. Ultimately, results of in vitro studies, as well as those from animal models, have to be validated by a human pilot study where the beneficial vs. deleterious effects can be monitored, eventually proceeding to clinical trials [92].

## 4. Safety in the Polyphenol World

Few studies focused on the safe use of polyphenols for disease prevention and treatment, but they are paramount to further promotion of their use in human health. In a previous study [45], we evaluated three structurally different flavonoid polyphenols: quercetin, as probably the most ubiquitous flavonoid in foods; monomeric flavanol epigallocatechin gallate (EGCG), found in certain seeds of leguminous plants, in grapes, and, more importantly, in tea [93,94]; and anthocyanin cyanidin-3-glycoside (C3G), found as a pigment in many red berries, particularly in blueberries [95,96]. These polyphenols showed moderate-to-high antioxidant capacity (1.69 ± 0.2, 4.29 ± 0.16, 1.02 ± 0.02 trolox equivalent antioxidant capacity [97], respectively), and several authors found that they could be detected in plasma at high concentrations after oral administration [98,99,100]. As an example, in a previous study, several tests were used to evaluate the brain accessibility of EGCG, C3G, and quercetin, their direct neurotoxicity and, their ability to protect brain neurons from oxidative damage [45]. As can be seen in Figure 1, all three studied polyphenols had a diverse profile regarding the evaluated parameters. In fact, while EGCG showed some neurotoxicity, it was also the molecule with lower BBB disruption and higher neuroprotective potential from oxidative-induced necrotic-like and apoptotic-like neuronal death. On the other hand, C3G had very low direct neurotoxicity, but it only protected neurons from necrosis and not from apoptosis; furthermore, it already showed some destabilization of BBB properties. Finally, and in spite of many promising studies [24,101,102], quercetin showed the worst profile, with significant BBB disruption, moderate-to-high neurotoxicity, and almost absent neuroprotection.

As stated before, few studies are focused on the toxicity or safety of consuming polyphenols. Of these, some studies state that the regular consumption of green tea and green-tea extracts seem to be safe [88], particularly in the form of the traditional infusion [91,103], but the use of concentrated extracts with doses of individual constituents, consumed in solid dosage form, may need additional study to guarantee their safe use [89]. The direct administration of moderate doses of resveratrol seem to be safe and cardioprotective [104]. Accordingly, the use of a resveratrol supplement (Longevinex) did not seem to elicit any adverse effects in an animal study, indicating beneficial effects and safe use [105]. At the same time, grape-seed extract seems to also be safe for healthy rats, even in high repeated doses, and revealed antioxidative and anti-inflammatory properties [90]. Conversely, the intraperitoneal administration of high doses of EGCG to diabetic mice may show cardiotoxicity [87].

The efficacy of polyphenols can be improved by their inclusion in new pharmaceutical formulations that direct them to specific targets, apparently avoiding adverse effects, as shown in a recent study using curcumin self-nanomicellizing solid dispersion directed toward Alzheimer’s treatment [106]. However, poor regulatory constrictions of commercial polyphenol supplements and nonpharmaceutical formulations are a concern for their safe use, as it is in the case of medicinal pomegranate products for cancer [107].

## 5. Conclusions

Polyphenols are promising molecules for the prevention and possibly the treatment of many human pathologies, namely, neurodegenerative diseases. However, like any pharmaceutical drug, they might show parallel adverse effects and/or toxicity, particularly due to the accumulation of high levels in the organism (Figure 2). More studies are needed to discern the relationship between consumption and safe plasma concentrations that are beneficial. Until further studies are performed, a more natural consumption of polyphenol-rich products, like fruits, vegetables, tea, and coffee, is the most beneficial, while the overconsumption of food supplements advertised as polyphenols or polyphenol-rich, but mostly still poorly controlled by regulatory agencies, may lead to higher circulating levels and higher risk for adverse effects. Nevertheless, such supplements may be a useful resource when dietary food sources are not available.

Modifying nutritional habits by the regular inclusion of polyphenol-rich fresh foods, like red fruits, tea, and natural juices, rather than with the excessive consumption of concentrated supplements, may have the most beneficial effect in long-term neuroprotection by increasing the organism’s adaptive natural defenses and modulating several pathological mechanisms involved in neurodegeneration.

## Figures and Tables

**Figure 1 antioxidants-09-00061-f001:**
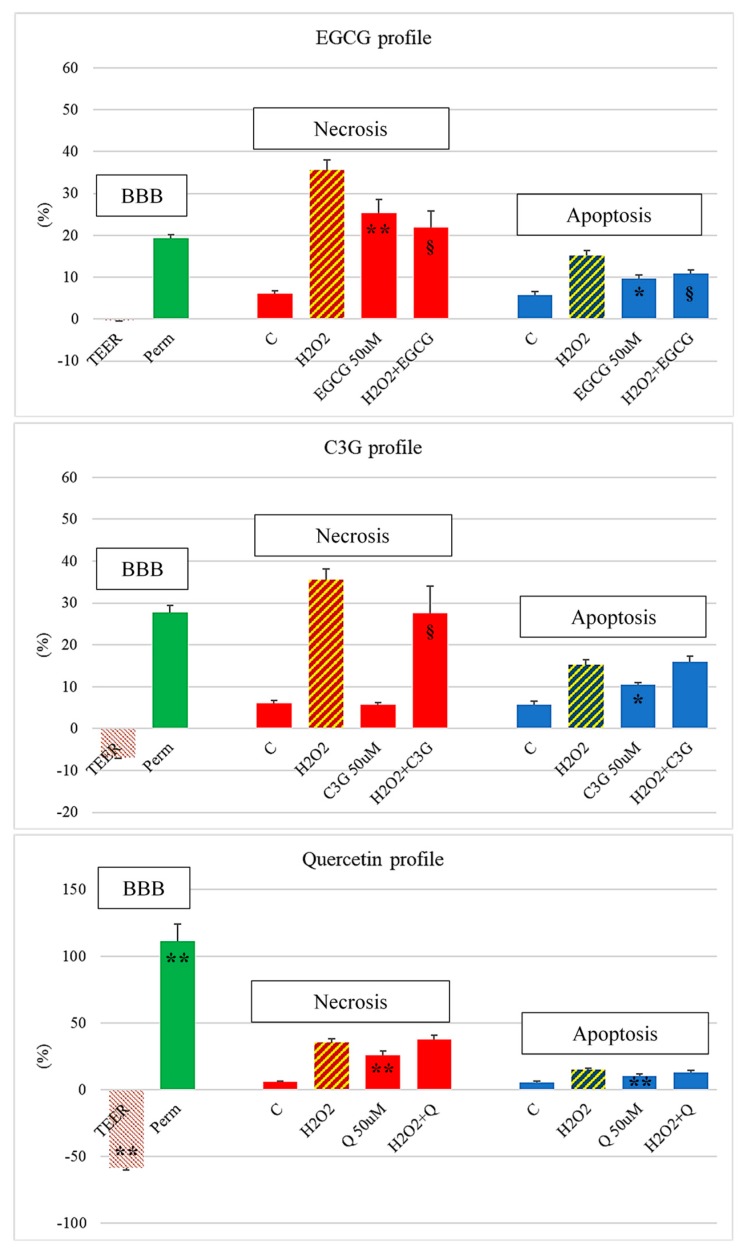
Neuroprotective and neurotoxic profile of epigallocatechin gallate (EGCG), cyanidin-3-glucoside (C3G), and quercetin (Q). Human brain microvascular endothelial cells (HBMEC) and rat-brain-neuron primary cultures were exposed to 50 µM of selected polyphenols for 24 h alone or after previous 24 h exposure to 150 µM H_2_O_2_. Controls were performed with respective culture media without H_2_O_2_ (C) or with H_2_O_2_**.** Following incubation, transendothelial electrical resistance and Na-fluorescein permeability were measured in HBMEC, and neurons were stained with propidium iodide and Hoechst 33342 dyes to evaluate necrosis-like and apoptosis-like cell death, respectively [45]. * *p* < 0.05 and ** *p* < 0.01 vs. control; § *p* < 0.05 vs. H_2_O_2_.

**Figure 2 antioxidants-09-00061-f002:**
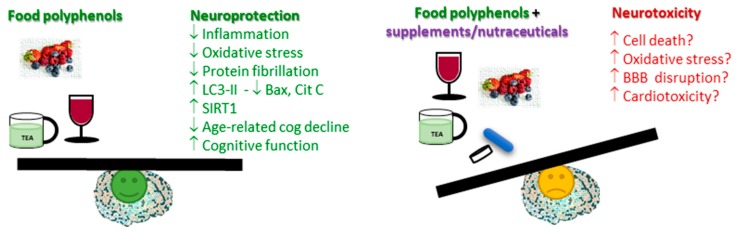
Summary of key mechanisms and actions from polyphenol neuroprotection, also highlighting possible safety concerns derived from polyphenol overconsumption.

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
