# Peer review of "Polyphenols from Food and Natural Products: Neuroprotection and Safety"

_antioxidants, 2020, doi:10.3390/antiox9010061_

Round 1

Reviewer 1 Report

The authors review the literature concerning the effect of polyphenols in neuroprotection, but also pose the question about the safety of an overconsumption of these compounds, which is a quite interesting and recent topic. Overall, the ms is clear, concise and informative. Only some comments are reported below.

Abstract: the main goal of the review should be described clearly Line 12-13: perhaps they are connected…(see line 58-60) Keywords: please follow alphabetical order and use semicolon for separation Line 31: I don’t know a plant species which lacks polyphenols -54-57 not clear for poor language Section 2: pretty good! Section 3: clear and very informative Table 1: for me a table with just a row and two coloumns is not necessary and data can be directly reported in the text. What could be interesting is to create a more articulate table to report the level of different polyphenols (maybe total or the most abundant e.g. anthocyanins in blueberry, etc…) showing their antioxidant potential. This would allow comparison among several food sources Line 230: three instead of tree Figure 1: it is not clear the origin of this dataset (publish papers? Which? Number 33? Unpublished? Please mention this). Also, the quality is very low, statistic is not applied. Please revise it When the authors will clarify the main goal of their review (I guess is the neuroprotection) it would be beneficial to add a table or a figure in which some key results (table) and a schematization of the effect of polyphenols in neuroprotection is schematized (figure). This is common in review articles and would be seriously beneficial for a reader.

Beside these, I have no othe comments and I think the paper could be accepted after the aforementioned changes will be undertaken.

Reviewer 2 Report

The review entitled "Polyphenols from Food Natural Products; Neuroprotection and Safety" presents an overview of some polyphenols as promising molecules for the prevention, and possibly for the treatment of many human pathologies, namely neurodegenerative disease. Also, the authors addressed the safety and tolerability issues of these compounds in the food supplements, being these last points the most relevant issue of the article.

However, please consider the following comments before final acceptance for publication in Antioxidants:

The text is easy to read, even so, a minor revision and spell-checking of the English language is suggested. The manuscript is well structured, presenting current references. However, I think that safety and tolerability issues should be improved with current references. The authors report several times, and in different parts of the article that there are "many studies" but they do not refer any citation to support these statements. If they had cited a current review article, this could be acceptable (although not preferred). On this way, must be added sources in the lines 39, 55, 84, 87, 114, 165, 167, 188, 191, 195-213 (the point 3.3 “Neurotoxicity vs. Neuroprotection” does not have a single reference, this is not acceptable), 216 (even if the study is by the authors themselves, it should be mentioned if it is already published), 219, 220. Lines 228-230 the authors refer to “As an example, in a previous study several tests were used to evaluate the brain accessibility of EGCG, C3G, and quercetin, as well as their direct neurotoxicity and, finally, their ability to protect brain neurons from an oxidative damage [59], ”the cited article (Pogacnik, L.; Rogelj, A.; Ulrih, N.P. Chemiluminescence Method for Evaluation of Antioxidant 450 Capacities of Different Invasive Knotweed Species. Anal Lett 2016, 49, 350-363) does not refer the mentioned studies - put the correct source. Line 230, I think the authors should mean "three" not "tree".

Reviewer 3 Report

The present study briefly explore the current state of knowledge regarding the potential use of polyphenols in neurodegenerative diseases. The paper is informative, but lack of systematic approach and is rather not comprehensive. However, if considered as a short review, it could be published. 

Major note:

The chapters should be better organized distinguishing in vitro, in vivo and human studies.

Just a few notes:

Line 39

I think the introduction is too vague regarding the dietary content of polyphenols. The authors report the results of 2 studies conducted in Mediterranean countries, but they don’t emphasize on the variability of dietary sources across countries and the geographical gradient we can observed in relation to the previous observation (in fact, 1 mg per day cannot be considered univocally consumed worldwide). Specifically, the authors may write the following: “major dietary sources of polyphenols may vary depending on the traditional diets adopted in various countries: thus, in Northern and Eastern European countries the main dietary sources of polyphenols are represented mostly by beverages, such as coffee and tea (PMID: 26081647 and PMID: 25280419), while in Southern European and Mediterranean countries important dietary sources may be nuts, olive oil, fruits and vegetables (PMID: 28276907 and PMID: 23332727).”

Line 154

There are actually 2 studies conducted on general population exploring the association between dietary polyphenols and depressive symptoms leading to similar results, including the potential role of phenolic acids (PMID: 27413131 and PMID: 29695122).

Round 2

Reviewer 3 Report

The authors revised the paper accordingly to the reviewer's suggestions.